# High-Fat Diet Augments Myocardial Inflammation and Cardiac Dysfunction in Arrhythmogenic Cardiomyopathy

**DOI:** 10.3390/nu16132087

**Published:** 2024-06-29

**Authors:** Ann M. Centner, Emily A. Shiel, Waleed Farra, Elisa N. Cannon, Maicon Landim-Vieira, Gloria Salazar, Stephen P. Chelko

**Affiliations:** 1Department of Biomedical Sciences, Florida State University College of Medicine, Tallahassee, FL 32306, USA; emily.shiel@med.fsu.edu (E.A.S.); waleed.farra@med.fsu.edu (W.F.); elisa.cannon@med.fsu.edu (E.N.C.); landimvieira.maicon@med.fsu.edu (M.L.-V.); 2Department of Health, Nutrition, and Food Sciences, College of Education, Health, and Human Science, Florida State University, Center for Advancing Exercise and Nutrition Research on Aging (CAENRA), Tallahassee, FL 32306, USA; gsalazar@fsu.edu; 3Division of Cardiology, Department of Medicine, Johns Hopkins University School of Medicine, Baltimore, MD 21218, USA

**Keywords:** arrhythmogenic cardiomyopathy, high-fat diet, Desmoglein-2, lipids, myocardium

## Abstract

Arrhythmogenic cardiomyopathy (ACM) is a familial heart disease characterized by cardiac dysfunction, arrhythmias, and myocardial inflammation. Exercise and stress can influence the disease’s progression. Thus, an investigation of whether a high-fat diet (HFD) contributes to ACM pathogenesis is warranted. In a robust ACM mouse model, 8-week-old Desmoglein-2 mutant (*Dsg2*^mut/mut^) mice were fed either an HFD or rodent chow for 8 weeks. Chow-fed wildtype (WT) mice served as controls. Echo- and electrocardiography images pre- and post-dietary intervention were obtained, and the lipid burden, inflammatory markers, and myocardial fibrosis were assessed at the study endpoint. HFD-fed *Dsg2*^mut/mut^ mice showed numerous P-wave perturbations, reduced R-amplitude, left ventricle (LV) remodeling, and reduced ejection fraction (%LVEF). Notable elevations in plasma high-density lipoprotein (HDL) were observed, which correlated with the %LVEF. The myocardial inflammatory adipokines, adiponectin (AdipoQ) and fibroblast growth factor-1, were substantially elevated in HFD-fed *Dsg2*^mut/mut^ mice, albeit no compounding effect was observed in cardiac fibrosis. The HFD not only potentiated cardiac dysfunction but additionally promoted adverse cardiac remodeling. Further investigation is warranted, particularly given elevated AdipoQ levels and the positive correlation of HDL with the %LVEF, which may suggest a protective effect. Altogether, the HFD worsened some, but not all, disease phenotypes in *Dsg2*^mut/mut^ mice. Notwithstanding, diet may be a modifiable environmental factor in ACM disease progression.

## 1. Introduction

Arrhythmogenic cardiomyopathy (ACM) is an inherited, nonischemic heart disease (NIHD) characterized by ventricular dysfunction, arrhythmias, and myocardial inflammation and fibrosis [1]. Despite its first description in the 1980–1990s [2,3], there is still no available drug to treat the underlying disease pathogenesis. ACM is often called “a disease of the cardiac desmosome” [4], as the first and second most prevalent pathogenic gene variants that give rise to ACM are the desmosomal genes Plakophilin-2 (*PKP2*) and Desmoglein-2 (*DSG2*) [5]. Interestingly, *PKP2* variants give rise to a right-ventricular-dominant form of ACM, whereas *DSG2* variants are left-ventricular dominant [6,7]. ACM is considered a disease with incomplete penetrance and variable expressivity, and external factors (e.g., stress, anxiety, and exercise) have been known to influence this disease’s manifestation and progression [4,8,9,10,11,12]. Whereas other lifestyle factors, such as diet, encompass an area that is largely unexplored in ACM progression and severity. On average, United States (US) adults consume a diet high in saturated fat, which is associated with arterial plaque formation, worsened systolic function, and elevated levels of circulating lipids [13,14,15,16]. Specifically, triglycerides (TGs) and elevated circulating cholesterols, including very low- and low-density lipoproteins (VLDLs and LDLs, respectively), are risk factors for cardiovascular disease (CVD), while high-density lipoprotein (HDL) cholesterol is regarded as heart-healthy, largely due to its ability to shuttle cholesterol for biliary excretion [17].

Poor diet is linked to obesity, an epidemic linked to a myriad of CVDs through dyslipidemia and the promotion of inflammatory pathways [18]. Given that fat is the most calorie-dense macronutrient, maintaining dietary fat within the recommended 20–35% daily consumption is of critical importance [19]. Astoundingly, Americans consume a diet consisting of, on average, 36% fat per day [20]. In addition, despite recommendations limiting saturated fat to 5–6% per day [19], the U.S. population consumes a diet enriched with fat far beyond the recommendations (11–12% per day) [21].

Saturated fat can elevate inflammatory cytokines [22,23], which were recently identified to drive myocardial injury, arrhythmias, and dysfunction in ACM [24,25]. Occurring in three out of four ACM patients, myocardial inflammation is a causative phenotype linked to premature death [2,26,27]. Recent work has demonstrated that the master regulator of inflammation and the innate immune response, NFκB, increases inflammatory cytokine and chemokine levels via the mobilization of immune cells to the hearts of ACM subjects [24,25]. Thus, disease expressivity and severity can be influenced by intrinsic (i.e., myocyte-mediated NFκB chemotaxis), cardiac-extrinsic (i.e., circulating factors), and environmental (i.e., stress and exercise) stimuli. As such, the topic of whether a high-fat diet (HFD) can influence disease progression in this familial heart disease warrants investigation.

A previous study reported that the HFD induced certain disease phenotypes in a Plakophilin-2 heterozygous (*Pkp2*^+/−^) mouse (e.g., 10-week-old mice received an HFD for 3 months) [28]. However, these mice were asymptomatic until an HFD was introduced, indicating that this environmental factor could be sufficient to induce disease onset in genotype-positive, phenotype-negative (G+/P−) ACM patients. Yet, whether an HFD is sufficient to influence disease progression in G+/P+ ACM subjects is unknown. Conversely, this body of work utilizes an ACM mouse model that develops cardiac dysfunction by early adolescence (<8 weeks old)—without exogenous stimuli (i.e., diet-induced). Considering elevated fat consumption in the general population and the cardiovascular effects of an HFD on healthy individuals as well as those with acquired heart diseases, we investigated the influence of an HFD in a preclinical animal model of ACM. In essence, whether an HFD exacerbates existing disease phenotypes (i.e., rather than inducing them [28]) has yet to be elucidated.

## 2. Materials and Methods

### 2.1. Animal Studies

Wildtype (WT) controls and homozygous Desmoglein-2 mutant (*Dsg2*^mut/mut^) mice [29] were enrolled in an 8-week dietary intervention study at 8 weeks of age. All the mice were housed in the Florida State University College of Medicine vivarium in a temperature- and humidity-controlled environment with a 12 h light–dark cycle with access to food and water ad libitum. All the experiments were approved under the Florida State University Animal Care and Use Committee (Protocol No. PROTO202000052). This study used both male and female mice, with equal sexes randomized to one of two diets: a purified HFD (60% kcal fat, 20% kcal carbohydrate, and 20% kcal protein; Research Diets, New Brunswick, NJ, USA, Cat. No. D12492) or standard rodent chow, non-purified diet (16.5% kcal fat, 56.8% kcal carbohydrate, 26.6% kcal protein; Lab Diets, Richmond, IN, USA, Cat. No. 5008).

### 2.2. Mouse Echo- and Electrocardiography

The cardiac function was assessed via echocardiography (Echo) using the Vevo F2 Visualsonic imaging system (FUJIFILM VisualSonics, Toronto, ON, Canada) with a UHF57x linear transducer (57–25 MHz) and 8-lead surface electrocardiography (ECG) using the iWork IX-BIO-SA Small Animal ECG System (Cat. No. TCS-100; iWorx Systems Inc., Dover, NH, USA). Echos and ECGs were performed on anesthetized mice (nose cone anesthesia, 1.5–2% isoflurane vaporized in 100% O_2_) prior to dietary intervention (8 weeks of age) and at study endpoint (16 weeks of age). The Echos were performed, measured, and analyzed according to the American Society of Echocardiography guidelines [30]. At least three short-axis (M-mode) and parasternal long-axis (B-mode) videos were acquired for each mouse and averaged, as previously described [29]. Five-minute ECG recordings were obtained via surface ECG telemetry electrodes following manufacturer’s protocol (iWork, IX-BIO8) and analyzed via iWork LabScribe v24 software following the construction of signal-averaged ECGs (SAECGs) from five-minute recordings. Following endpoint functional studies, the mice were euthanized, and tissues were collected and either flash-frozen in liquid nitrogen or stored in 10% buffered formalin. The flash-frozen samples were transferred to −80 °C, and fixed samples were processed and embedded in paraffin blocks as described below.

### 2.3. Plasma Marker Assessment

Blood was collected in lavender EDTA-coated tubes (BD Microtainer, Franklin Lakes, NJ, USA; Cat. No. 365974) and centrifuged at 2000× *g* for 10 min at 4 °C: the plasma was then transferred to Eppendorf tubes and stored at −80 °C until downstream use. The plasma samples were diluted (1:2–1:4) in 1X PBS, and then, 100 µL of diluted plasma samples were individually added to Lipid Panel Plus Reagent Discs (McKesson, Irving, TX, USA; Cat. No. 07P0205) and read in a Chemistry Analyzer Piccolo Xpress (Abbott, Chicago, IL, USA). Due to the variations in the sample concentrations and the sensitivity of the Piccolo Xpress analyzer, certain samples were re-run at different dilutions (i.e., 1:2–1:4) to be accurately assessed. Each sample was read individually using disc-specific calibration data via individual bar codes printed on each disc. Additionally, even when a sample is read by the machine, the output may not provide values for all markers. This is often due to undetectable levels of certain types of cholesterol.

### 2.4. Cardiac Fibrosis

The formalin-fixed hearts underwent a series of ethanol dehydration steps and were paraffin-embedded, sectioned (5–10 µm), mounted onto charged TissueTek frosted slides (Sakura, Osaka, Japan; Cat. No. 9035), and then deparaffinized, as previously described [25]. The slides were submerged in sequential order with water washes between each step as follows: (i) Bouin’s (Sigma-Aldrich, St. Louis, MO, USA; Cat. No. HT10132) at 56 °C for 15 min, (ii) Weigert’s Iron Hematoxylin (Sigma-Aldrich, Cat. No. HT1079) for 2 min, (iii) Biebrich Scarlet-Acid Fuchsin (Sigma-Aldrich, Cat. No. HT15) for 5 min, (iv) phosphotungstic (Sigma-Aldrich, Cat. No. HT152) and phosphomolybdic acid (Sigma-Aldrich, Cat. No. HT153) for 2 min, (v) Aniline Blue (Sigma-Aldrich, Cat. No. HT154) for 5 min, then (vi) 1% acetic acid for 2 min. The slides were then dehydrated through a series of ethanol steps and left to air dry, and then, DPX mountant media (Sigma-Aldrich, Cat. No. 06522) and a coverslip were added. The slides were imaged by light microscopy using a BZ-X800 Keyence Microscope (Leica Microsystems, Deerfield, IL, USA). Myocardial fibrosis was analyzed as the percentage of the sum of all fibrotic lesions divided by the total myocardium.

### 2.5. Cardiac Cytokine Arrays

Frozen hearts were lysed in a RIPA buffer (ThermoFisher, Waltham, MA, USA; Cat. No. 89900) with 1:100 protease (Sigma-Aldrich, Cat. No. P8430) and phosphatase inhibitor cocktails (Sigma-Aldrich, Cat. No. P0044). The protein levels were standardized using a Pierce BCA protein assay kit (ThermoFisher, Cat. No. 23227). Mouse Proteome Profiler XL Cytokine Arrays (R&D Systems, Minneapolis, MN, USA; Cat. No. ARY028) were blocked for 1 h at room temperature before the membranes were incubated at 4 °C with 200 µg of protein overnight. The following day, the membranes underwent the following steps with three water washes (10 min each) between each step: (i) incubated for 1 h at room temperature with antibody detection cocktail, (ii) incubated with 1:2000 streptavidin horseradish peroxidase for 30 min at room temperature, and then (iii) SuperSignal West Pico PLUS chemiluminescence substrate (ThermoFisher, Cat. No. 34577) was added before imaging on an Azure Biosystems 400 imager (Azure Biosystems Inc., Dublin, CA, USA). The images were quantified using the Ideal Eyes Systems, Inc. (Bountiful, UT, USA) QuickSpots software (version 25.6.0.1).

### 2.6. Statistical Analysis

The data are presented as the mean ± SEM with *p* ≤ 0.05 considered statistically significant. Additional statistical information is included in each figure legend. GraphPad PRISM was used for all statistical computations and graph preparations. Differences between groups and treatments were assessed with one- and two-way ANOVAs with Tukey’s post hoc analysis as appropriate, and normal distributions were assumed. Pearson’s correlation coefficient (r) was used to determine correlations between the Echo parameters and the circulating lipids.

## 3. Results

### 3.1. Impaired Cardiac Function and ECG Anomalies Following High-Fat-Diet Exposure in Dsg2^mut/mut^ Mice

As aforementioned, this study was designed to fill a gap in our knowledge of environmental factors that impact ACM, namely, how an HFD affects disease progression in ACM. The choice of HFD for this project was one of the most common HFDs, if not the most common HFD, for basic scientists. Sold for over 20 years by the largest supplier of rodent diets, Research Diets, the 60% kcal fat diet reduces the time of animal housing and the study duration without sacrificing study quality [31]. Specifically, in comparison to the 45% kcal fat diet offered by Research Diets, the metabolic response differences are very small [31].

Thus, during this dietary modification period, which occurred from 8 to 16 weeks of age, the *Dsg2*^mut/mut^ mice received standard rodent chow or an HFD (60% kcal fat). WT mice maintained on the same standard rodent chow served as the study control group. As expected, the HFD increased body weight of the *Dsg2*^mut/mut^ mice, as seen in Appendix A.

The most notable changes in cardiac function are portrayed in Figure 1, while additional echocardiographic data are compiled in Table 1. As assessed by the percent left ventricular ejection fraction (%LVEF), compared to WT controls (Figure 1A,B), cardiac dysfunction was similar in both *Dsg2*^mut/mut^ cohorts prior to dietary intervention. These data ensured uniformity prior to study initiation and enabled us to assess whether an HFD can exacerbate existing disease phenotypes. While both chow- and HFD-fed 16-week-old *Dsg2*^mut/mut^ mice continued to display reduced cardiac function (both *p* < 0.0001 compared to WT controls), further impairment in cardiac function was observed in HFD-fed 16-week-old *Dsg2*^mut/mut^ mice (*p* < 0.0011 compared to *Dsg2*^mut/mut^ mice fed chow; Figure 1A,B, Table 1). This finding indicated that exposure to a diet high in fat is sufficient to potentiate LV dysfunction in ACM mice.

A variety of ECG anomalies were identified at 16 weeks of age (Figure 1C, Table 1). Specifically, the P-wave duration was shortened in chow- (*p* = 0.0021) and HFD-fed (*p* = 0.0016) *Dsg2*^mut/mut^ mice compared to WT controls (Figure 1D,E). Additionally, the P-wave amplitude was reduced in chow- (*p* = 0.2361) and HFD-fed (*p* = 0.0043) *Dsg2*^mut/mut^ mice compared to WT controls (Figure 1D,E). Similar to our previous findings [24], the S-amplitude trended lower in *Dsg2*^mut/mut^ mice fed standard rodent chow, yet *Dsg2*^mut/mut^ fed an HFD displayed near similar values in S-amplitude compared to WT controls (Figure 1F). In addition, as observed in Figure 1C and Table 1, the R-amplitude was diminished in both *Dsg2*^mut/mut^ groups compared to the WT controls. In light of the further decline in cardiac function in HFD-fed *Dsg2*^mut/mut^ mice and considering that a reduced %EF is a major Task Force criteria for ACM [32], we performed correlation analyses between Echo parameters for this cohort (Figure 2A,B). The echocardiographic analyses demonstrated increased LV mass (LVM) with increased thinning of the LV wall (RWT; Figure 2A) in HFD-fed *Dsg2*^mut/mut^ mouse hearts. And a strong, positive correlation between %LVEF and RWT (r = 0.77) and %LVEF and LV fractional shortening (%LVFS, r = 1.0) was observed (Figure 2C,D). In short, the %LVEF, RWT, and %LVFS were all drastically lower in *Dsg2*^mut/mut^ mice fed an HFD than those in all cohorts (*p* < 0.01). Thus, our results demonstrate that an HFD not only led to further cardiac dysfunction but was an additional driver of cardiac remodeling.

### 3.2. HFD-Induced Elevations in Plasma Lipids in Dsg2^mut/mut^ Mice

Next, we measured plasma lipids and whether they correlated with cardiac functional parameters (Figure 3A) or with additional circulating factors (Figure 3B)—an investigative outcome that could lead to dietary recommendations in patients with ACM. The HFD elevated the circulating levels of total cholesterol (*p* < 0.001 compared to both cohorts, TC, Figure 3C). This finding was novel, as a previous report in mice with the same C57BL/6 background as the *Dsg2*^mut/mut^ mice had a trend toward a higher TC with 16 weeks of an HFD modification [33]. The majority of the cholesterol was in LDL in HFD-fed *Dsg2*^mut/mut^ mice (Figure 3D), which was undetected in both cohorts on chow diet. However, the HDL was increased in HFD-fed *Dsg2*^mut/mut^ mice compared to WT mice (*p* < 0.0348) and *Dsg2*^mut/mut^ mice (*p* < 0.0060) fed chow (Figure 3E). Of interest, the only lipid that correlated with %LVEF was HDL (r = 0.83), indicating that the HDL may act to preserve cardiac function in this ACM model. Surprisingly, TG and VLDL, strong CVD risk factors [34], were elevated in *Dsg2*^mut/mut^ mice fed standard rodent chow but were similar between WT mice fed chow and *Dsg2*^mut/mut^ mice fed an HFD (Appendix A). In line with the observed increase in LDL, the nHDL-C was additionally elevated in *Dsg2*^mut/mut^ mice fed an HFD, while both *Dsg2*^mut/mut^ cohorts displayed a higher TC/HDL ratio (Appendix A). Lastly, the liver enzymes, ALT and AST, are both associated with organ-specific and systemic inflammation [7], yet they were unchanged between cohorts, as was glucose (Appendix A). Overall, our plasma analyses indicated a paucity of correlation between circulating lipids and cardiac function. Lastly, and most surprisingly, it is worth repeating that a potential cardioprotective mechanism involves circulating HDL levels in response to an elevated dietary fat consumption in ACM mice.

### 3.3. HFD Consuption Elevated Myocardial Inflammatory Cytokines in Dsg2^mut/mut^ Mice

Continuing our investigation into potential instigators that could be responsible for the adverse effect on cardiac function in HFD-fed *Dsg2*^mut/mut^ mice, we assessed myocardial fibrosis (Figure 4A). As previously reported by our group, healthy, age-matched WT hearts have null-to-minimal fibrosis, while *Dsg2*^mut/mut^ mice at 16 weeks have severe fibrotic scarring [24]. Although similar findings were noted in chow-fed *Dsg2*^mut/mut^ mice compared to WT controls, our results indicated that the HFD did not have a compounding effect on cardiac fibrosis (Figure 4A,B).

Since the fibrotic burden was not elevated in response to this dietary modification, we turned our attention to cardiac inflammation, a key pathological signature of ACM [2,26,27] that has been previously reported in this mouse model [24,25]. We found numerous markers of cardiac inflammation that were elevated with an HFD compared to *Dsg2*^mut/mut^ and WT mice fed chow (Figure 4C,D, Appendix A). Specifically, in *Dsg2*^mut/mut^ mice fed chow, cardiac elevations in periostin (POSTN; two- to threefold), tumor necrosis factor alpha (TNFα; three- to fourfold), and pentraxin-2 (PTX2; three- to fourfold) and -3 (PTX3; two- to threefold) were observed (Appendix A) compared to chow-fed WT mice. When HFD-fed *Dsg2*^mut/mut^ mice were compared against WT controls, matrix metalloproteinases-2/3/9 (MMP2, two- to threefold; MMP3, three- to fourfold; and MMP9, three- to fourfold; respectively), osteopontin (OPN, two- to threefold), lipocalin-2 (LCN2, two- to threefold), plasminogen activator inhibitor-1 (PAI-1, three- to fourfold), myeloperoxidase (MPO, two- to threefold), resistin (RETN, two- to threefold), C-X-3C motif chemokine ligand-1 (CX3CL1, two- to threefold), C-C motif chemokine ligand-21 (CCL21; two- to threefold), and PTX2 (two- to threefold) were all elevated (Appendix A). Thus, a myriad of myocardial inflammatory markers were elevated in response to ACM mice fed an HFD (Appendix A).

Lastly, fold changes do not take into account the total protein levels of inflammatory markers, as these are normalized to WT controls. Thus, we additionally assayed these markers as picograms per milliliter (pg/mL), which highlighted even more findings (presented in Figure 5). This additional analysis was performed considering the physiological role and relevance that each cytokine and/or chemokine imparts (Figure 5). Thus, the inflammatory markers were categorized into their respective functional role, as previously described [35]. For cytokine ligands, adiponectin (AdipoQ) was the only marker elevated in cardiac tissue from *Dsg2*^mut/mut^ mice fed an HFD compared to both *Dsg2*^mut/mut^ and WT mice fed a chow diet (Figure 5A). When grouped by their role in chemotaxis, the most elevated levels of chemokines were observed for CCL21 and CX3CL1 in *Dsg2*^mut/mut^ mice fed an HFD compared to both chow-fed cohorts (Figure 5B). The inflammatory receptors and those involved in cascade signaling, CD93 and soluble ICAM-1 (sICAM-1), were elevated in HFD-fed *Dsg2*^mut/mut^ mice compared to the other two cohorts (Figure 5C). Additionally, HFD-fed *Dsg2*^mut/mut^ mice harbored higher levels of four markers involved in enzymatic activity, including cluster of differentiation 142 (CD142, aka thromboplastin), *C*-reactive protein (CRP), PTX2, and PAI-1 (Figure 5D). Within the hormone and growth factor group, *Dsg2*^mut/mut^ mice fed an HFD (compared to *Dsg2*^mut/mut^ and WT mice fed chow) showed the highest levels of fibroblast growth factor-1 (FGF1; >18,000 pg/mL) compared to any other cytokine (Figure 5E,F), an unexpected finding. Lastly, compared to both chow-fed cohorts, *Dsg2*^mut/mut^ mice fed an HFD had potentiated levels of several markers, including fetuin A (AHSG), endoglin (CD105), and endostatin (Figure 5F). All of these inflammatory markers are known to augment cardiac remodeling and can additionally alter electrophysiological conduction via autocrine/paracrine signaling of neighboring ion channels [36,37].

## 4. Discussion

### 4.1. HFD and Cholesterol: Their Role in Myocardial Inflammation

As previously described, in a complex and intricate network of signaling events, an HFD elicits multiorgan changes, driving systemic inflammation [38]. The HFD elevates endotoxins and free fatty acids, increases NFκB signaling and oxidative stress, and triggers the activation of M1 macrophages [38]. This leads to the production of an abundance of inflammatory cytokines that can induce systemic inflammation and promote a myriad of diseases [38]. In atherosclerosis and coronary heart disease, an HFD elevates IL-6, *C*-reactive protein (CRP), and TNF-α preceding endothelial dysfunction and nitric oxide depletion. The HFD’s role in nonischemic heart diseases, such as ACM, is largely unknown, particularly regarding whether it can exacerbate existing ACM phenotypes. Considering prior studies demonstrating the role of NFκB signaling and oxidative stress in ACM [12,24], we suspected that an HFD would augment these existing phenotypes.

The HFD upregulated the TC, most of which was LDL, while the HDL was also increased. A higher baseline HDL and undetectable levels of LDL with the chow diet were expected, as mice preferentially carry cholesterol on HDL in contrast to humans, who preferentially carry cholesterol on LDL [39]. As such, LDL has long been considered a risk factor and therapeutic target for CVDs [40]. Furthermore, elevated levels of oxidized LDL (oxLDL) have previously been observed in ACM patients and were associated with cardiac dysfunction, structural remodeling, and ECG abnormalities [28]. Through additional sophisticated iPSC and animal studies, Sommariva E. and colleagues demonstrated that these phenotypes were the result of PPARγ-dependent cardiac adipogenesis [28], an additional pathway implicated by us [41].

Here, the positive association of HDL with the %LVEF in ACM mice fed an HFD led us to believe that HDL may exert protective effects. Widely regarded for its role in reverse cholesterol transport, HDL also has anti-oxidant and anti-apoptotic effects [17]. However, most recently, the oxidation of HDL has been associated with atrial fibrillation (AFib) [42], while high HDL has been associated with dementia (>90 mg/dL) [43] and, in male patients, with hypertension, a CV risk factor (>80 mg/dL) [44]. Further investigation is warranted, especially considering the disparate outcomes in human studies, where HDL showed a U-shaped association, in which low and high levels are associated with poor outcomes [45]. In particular, high levels of HDL either showed no cardiovascular benefits [45] or were associated with increased CVD-related mortality at severely elevated levels, such as ≥90 mg/dL [46] or ≥116 mg/dL [47], from two independent studies. Overall, attention to the oxidation status and plasma levels in relation to other health indices is vital to delineate the protective or adverse effects of HDL cholesterol.

### 4.2. Inflammation-Induced Cardiac Remodeling

CD93 is primarily expressed by endothelial cells, which line the endocardium [48], thus making this a suitable environment for CD93 storage and release. Additionally, elevated levels of circulating CD93 are associated with cardiac mortality and AFib [49]. CD93 and sICAM-1 were elevated in HFD-fed *Dsg2*^mut/mut^ mice compared against the other two cohorts (threefold and twofold, respectively). Considering that both CD93 and sICAM-1 are known promoters of atherosclerosis, it is not surprising that these inflammatory markers would additionally compound existing CVDs [50,51]. In addition, in mice with heart failure, sICAM-1 mediates leukocyte infiltration, leading to LV cardiac remodeling and subsequent dysfunction and fibrosis [52]. Conversely, we saw cardiac dysfunction in the absence of heightened fibrosis in HFD-fed *Dsg2*^mut/mut^ mice. This could be a consequence of MMP overexpression. CRP levels were also elevated with an HFD in *Dsg2*^mut/mut^ mice. CRP is secreted by immune cells and ACM myocytes in response to acute inflammation [53], and decreased levels of CRP are associated with QT shortening [54]. It did come as a surprise to see elevated PXT2 levels with HFD, as this cytokine is involved in antifibrotic and anti-inflammatory signaling pathways [55]. However, PXT2 was not associated with other HFD-induced upregulated cytokines in ACM subjects, as indicated in Appendix A. Thus, its increase may be a protective mechanism against myocardial fibrosis.

Alternatively, it comes as no surprise that AdipoQ was high in all mice compared to other markers, as this circulating hormone is the most abundant adipokine in circulation [56], with both atheroprotective and atheropromoting effects reported by us previously [57]. Observationally, increased plasma levels are associated with a heightened risk of heart failure, AFib, aortic valve stenosis, and myocardial infarction (MI) [58], while another study found that reduced AdipoQ levels were associated with AFib [59]. While AdipoQ can be secreted by a wide array of cell types, including aortic vascular smooth muscle cells [60], the primary cell type that secretes this adipokine are adipocytes [61]. Here, the observation of elevated myocardial levels of AdipoQ in response to the HFD suggests that this adipokine may be involved in ACM fibrotic scarring and/or adipogenesis, hallmarks of ACM pathology [62,63]. Overall, while these inflammatory marker elevations do not appear to further myocardial fibrosis, they may be involved in cardiac remodeling.

A reduced level of fetuin-A is linked to LV remodeling, increased mortality, post-MI, and ECG alterations, including incomplete ST segment resolution [64]. Thus, elevated AHSG with the HFD in our ACM mouse model may be protective. COL18A1, which was elevated with the HFD, has been associated with right ventricle dysfunction [65]. PAI-1, a coagulation factor increased with an HFD, is associated with prevalent and incident AFib [66]. CCL21 is also associated with detrimental effects including associations with all-cause mortality, myocardial inflammation and fibrosis, and chronic heart failure [67,68]. Lastly, the prevention of CCL21 binding to CCR2 prevents myocyte action potential (AP) prolongation [69].

CX3CL1 was an additional inflammatory marker elevated by the HFD. CX3CL1 (aka, fractalkine) is associated with AFib [70], promotes leukocyte recruitment and monocyte-adipocyte adhesion [71], and upregulates MMPs and ICAM [72]. CX3CL1 affects numerous cell-signaling pathways, including NFĸB and Wnt/β-catenin [73]. As P-wave abnormalities are indicative of the established involvement of canonical Wnt signaling in the control of AV junction electrophysiology [74], it is conceivable that CX3CL1 could contribute to P-wave alterations.

Also increased by the HFD was CD105, which has both anti- and profibrotic effects [75]. When CD105 activity is blunted, cardiac fibrosis is also blunted and associated with reduced heart failure mortality [76]. CD105 is a member of the transforming growth factor-β (TGF-β) receptor complex and promotes endothelial cell proliferation and migration [77]. On the other hand, CD105 downregulates TGF-β and NFκB signaling to exert antifibrotic effects [78]. Thus, our findings may implicate elevated CD105 levels as an antifibrotic compensatory mechanism as a potential cardioprotective cytokine.

FGF1 levels were substantially increased in *Dsg2*^mut/mut^ mice fed an HFD; in fact, FGF1 was the most elevated cytokine in this cohort. Given FGF1 is involved in fibroblast and endothelial cell growth during cardiac cell repair [79], it affords a plausible explanation for the stark potentiation of FGF1. Also elevated in mice subjected to an HFD, RETN is secreted by adipocytes, promoting inflammation in an NFκB-dependent mechanism [80]. This is particularly interesting, considering NFκB signaling is implicated in ACM pathogenesis and could serve as a therapeutic target in the treatment of ACM [24,25,81]. Increased MMPs (-2, -3, and -9) suggest extracellular matrix (ECM) remodeling and fibrosis, key ACM cardiac alterations, which can be further investigated in vivo and in vitro. While we did not see increased cardiac fibrosis in HFD-fed *Dsg2*^mut/mut^ mice, it is worth noting that MMP-induced ECM remodeling increases vascular permeability and, thus, access to infiltrating immune cells in ACM hearts [82], a finding that we recently reported was the result of infiltrating CCR2^+^ macrophages [25].

### 4.3. HFD-Induced ECG Alterations and Cardiac Dysfunction

Although the P-amplitude was reduced in both *Dsg2*^mut/mut^ cohorts, only those mice on the HFD showed a significant reduction compared to WT controls. This is particularly interesting given that our prior studies demonstrated no difference in the P-amplitude between WT and *Dsg2*^mut/mut^ mice [24,29], except in exercised *Dsg2*^mut/mut^ mice [12]. Thus, demonstrating that both environmental factors (i.e., diet and exercise) can influence atrial depolarization.

Although originally termed arrhythmogenic right ventricular cardiomyopathy (ARVC), considering the prevalence of right-ventricular-dominant phenotypes, now this dreadful disease can also manifest as biventricular disease or as left-dominant ACM [83]. Left-dominant disease is more prevalent in subjects harboring a *DSG2* pathogenic variant; and thus, in addition to the observed reductions in the %LVEF in HFD-fed *Dsg2*^mut/mut^ mice, there were other LV echocardiographic indices worth noting. In particular, *Dsg2*^mut/mut^ + HFD mice showed increased thinning of the LV wall (i.e., relative wall thickness [RWT]), in conjunction with increased LV mass (LVM) and severely reduced LV fractional shortening (%LVFS; Table 1). Both indices, RWT and LVM, were far worse compared to WT + chow as well as HFD-fed *Dsg2*^mut/mut^ mice (Table 1). The further and stark decline in the %LVFS in HFD-fed *Dsg2*^mut/mut^ mice compared to all cohorts is of clinical importance, given that LV dysfunction is traditionally defined as having a %LVEF of less than 50% and a %LVFS of less than 28% [84,85] in patients with heart disease. However, our findings indicate that the %LVFS may be a diet-induced functional parameter worth monitoring in ACM subjects, given the fact that HFD-fed *Dsg2*^mut/mut^ mice failed to achieve an LVFS of 28% or more.

## 5. Conclusions

Originally believed to be a congenital heart disease (i.e., ARVD), five decades of research has now uncovered that ACM is a progressive and primary NIHD arising from, chiefly, pathogenic desmosomal variants. Environmental Factors: Numerous studies have demonstrated the following: (i) Exercise worsens disease progression and can induce fatal arrhythmias, yet only secondary to the underlying antecedent (i.e., pathogenic variant). (ii) Recent work even demonstrated the impact of psychosocial stress on potentiating the arrhythmic burden in both mouse models of ACM and patients with ACM. Thus, (iii) whether an HFD could exacerbate disease phenotypes is of monumental importance, as this is a modifiable environmental factor. Our findings further advance the field of ACM, by demonstrating that an HFD further increases the circulating TC, potentiating cardiac dysfunction and myocardial inflammation, in the presence of the primary root cause (i.e., *Dsg2*-mutation). Thus, while an HFD is a well-known contributor to cholesterol burden, atherosclerosis, and CVDs, it is not the precursory trigger in cardiac dysfunction in ACM mice. However, it should be considered an environmental factor that advances disease progression.

## Figures and Tables

**Figure 1 nutrients-16-02087-f001:**
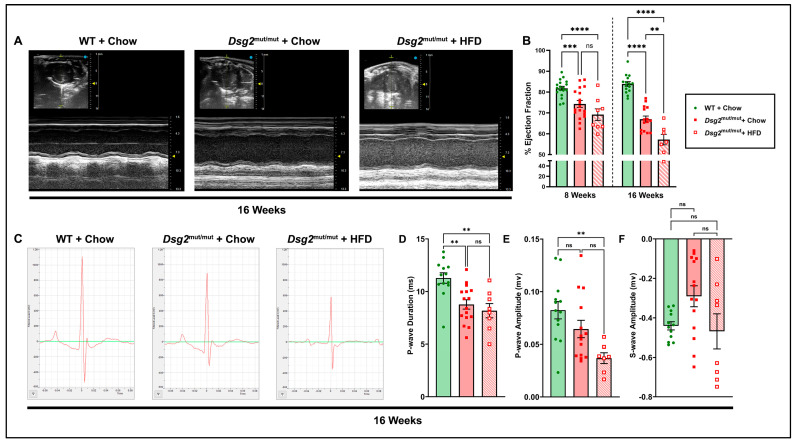
Echo- and electrocardiographic alterations in *Dsg2*^mut/mut^ mice fed HFD. (**A**) Representative short-axis m-mode echocardiograms (Echos) from 16-week-old WT mice fed chow (WT + chow) and *Dsg2*^mut/mut^ mice fed chow (*Dsg2*^mut/mut^ + chow) or HFD (*Dsg2*^mut/mut^ + HFD). (**B**) Cardiac function assessed as percent left ventricular ejection fraction (%LVEF), prior to diet intervention (8 weeks of age) and at study endpoint (16 weeks of age). (**C**) Representative signal-averaged electrocardiograms (SAECGs) from 16-week-old mice and indices of interest altered by an HFD, including (**D**) P-wave duration, (**E**) P-wave amplitude, and (**F**) S-wave amplitude. ms, millisecond; mv, millivolt. Data presented as mean ± SEM; *n* ≥ 7 mice/cohort/parameter; ns, not significant; ** *p* ≤ 0.01, *** *p* ≤ 0.001, **** *p* ≤ 0.0001 via one-way ANOVA with Tukey’s post hoc.

**Figure 2 nutrients-16-02087-f002:**
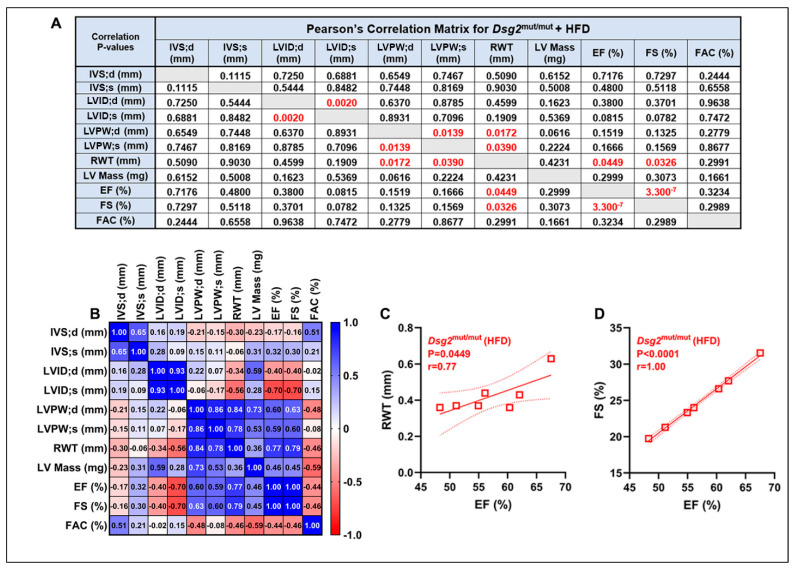
Echocardiographic correlations in *Dsg2*^mut/mut^ mice. (**A**,**B**) Pearson’s correlation matrix from echocardiographic indices taken from 16-week-old HFD-fed *Dsg2*^mut/mut^ mice (*n* ≥ 7 mice/parameter). Pearson’s correlation (**A**) *p*-values and (**B**) r-values. Positive correlations, blue; negative correlations, red. Note: (**C**) relative wall thickness (RWT) and (**D**) LV fractional shortening (%LVFS) demonstrated a strong correlation with %LVEF via Pearson’s r correlation analysis. *n* ≥ 7 mice/cohort/parameter. Square boxes, each HFD-fed *Dsg2*^mut/mut^ mouse’s individual data points; solid and dashed lines, the linear regression line (i.e., slope) and 95% confidence interval, respectively.

**Figure 3 nutrients-16-02087-f003:**
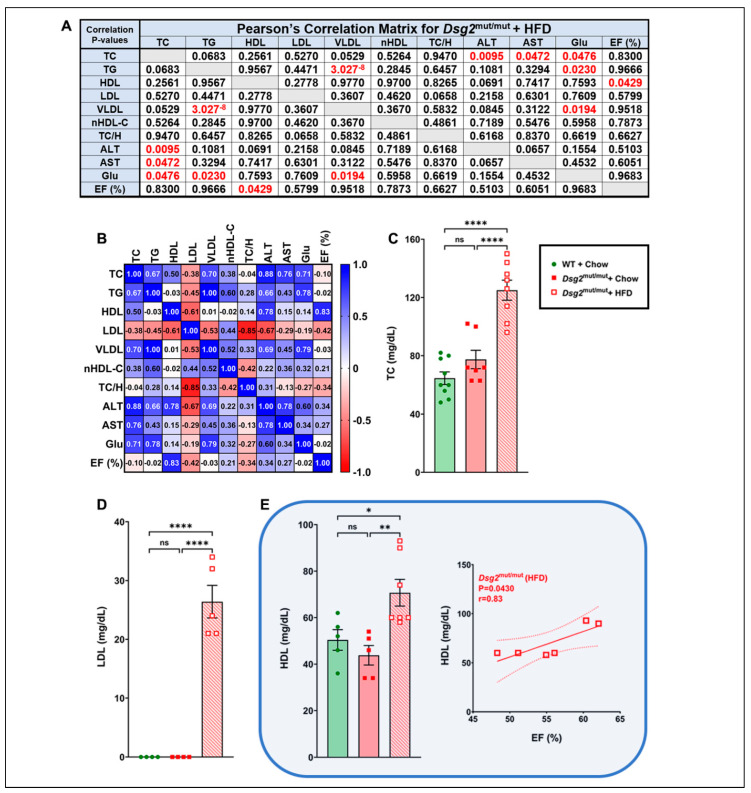
Lipid perturbations and ejection fraction correlations in *Dsg2*^mut/mut^ mice. (**A**,**B**) Pearson’s correlation matrix circulating for lipids and plasma markers from 16-week-old HFD-fed *Dsg2*^mut/mut^ mice (*n* ≥ 4 mice/parameter) in relation to %LVEF. Pearson’s correlation (**A**) *p*-values and (**B**) r-values. Positive correlations, blue; negative correlations, red. For (**A**), any correlation deemed significant via * *p* ≤ 0.05. (**C**–**E**) HFD-fed augmented the levels of circulating TC (mg/dL), LDL (mg/dL), and HDL (mg/dL), respectively, in *Dsg2*^mut/mut^ mice compared to all cohorts. (**E**) Of note, HDL levels correlated with increased %LVEF (r = 0.83). Square boxes, each HFD-fed *Dsg2*^mut/mut^ mouse’s individual data points; solid and dashed lines, the linear regression line (i.e., slope) and 95% confidence interval, respectively. For (**C**–**E**), data presented as mean ± SEM; ns, not significant; * *p* ≤ 0.05, ** *p* ≤ 0.01, and **** *p* ≤ 0.0001 via one-way ANOVA with Tukey’s post hoc. *N* ≥ 4mice/cohort/parameter. TC, total cholesterol; TG, triglycerides; HDL, LDL, and VLDL, high-, low-, and very low-density lipoproteins, respectively; nHDL-c, non-HDL cholesterol; TC/H, TC/HDL; ALT, alanine aminotransferase; AST, aspartate aminotransferase; Glu, glucose.

**Figure 4 nutrients-16-02087-f004:**
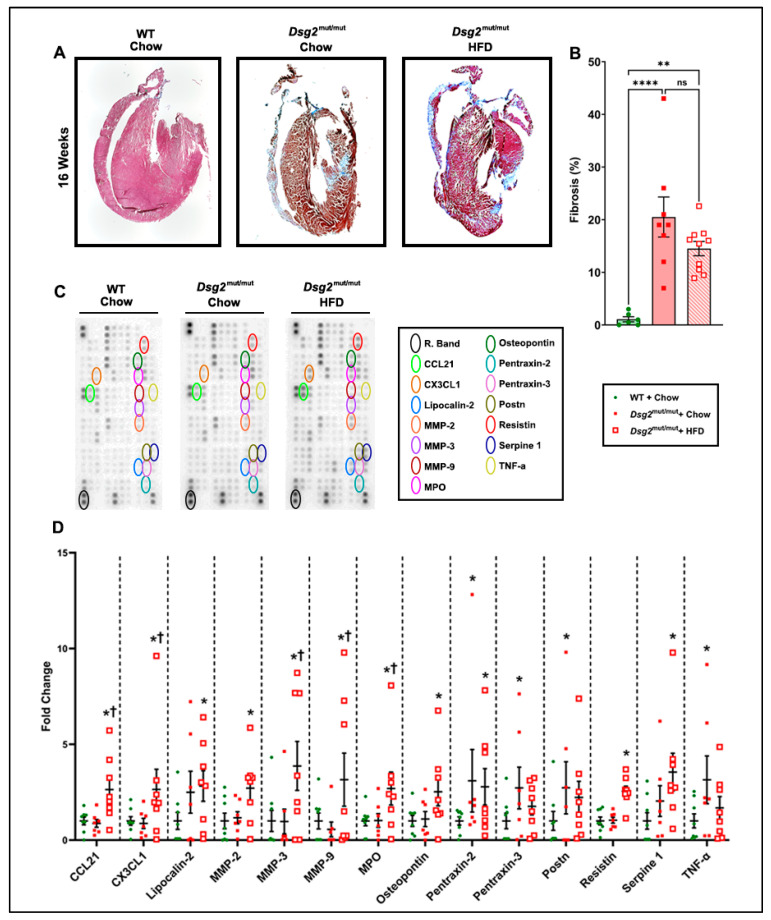
HFD-induced exacerbation of inflammatory cytokines in *Dsg2*^mut/mut^ mice. (**A**) Representative Masson’s trichrome immunostained hearts from 16-week-old mice. (**B**) Note the inability of an HFD to induce further myocardial fibrosis in *Dsg2*^mut/mut^ mice. ns, not significant; ** *p* ≤ 0.01, **** *p* ≤ 0.0001 via one-way ANOVA with Tukey’s post hoc. (**C**) Representative cytokine arrays and (**D**) grouped data from cardiac lysates from 16-week-old mice. Reference bands (R. Band). Data presented as mean ± SEM as fold change to WT controls; * *p* ≤ 0.05 any cohort vs. WT + chow; ^†^ *p* ≤ 0.05 *Dsg2*^mut/mut^ + HFD vs. *Dsg2*^mut/mut^ + chow. *n* ≥ 7 mice/cohort/parameter.

**Figure 5 nutrients-16-02087-f005:**
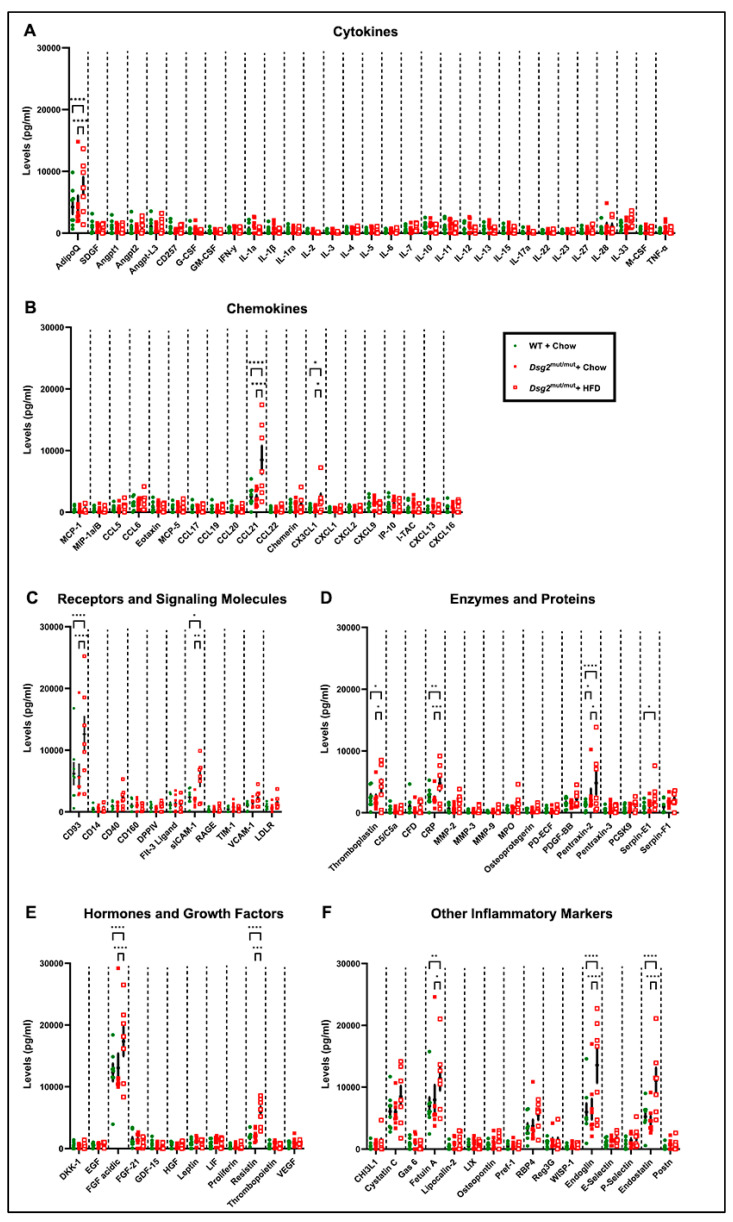
Myocardial inflammatory markers grouped by physiological role in cell signaling. (**A**–**F**) Levels of cardiac inflammatory cytokines (expressed as pg/mL) from 16-week-old mice grouped by role in inflammation. Data presented as mean ± SEM; *n* ≥ 7 mice/cohort/parameter; * *p* ≤ 0.05, ** *p* ≤ 0.01, *** *p* ≤ 0.001, and **** *p* ≤ 0.0001 via two-way ANOVA with Tukey’s post hoc.

**Table 1 nutrients-16-02087-t001:** Morphometric, echo-, and electrocardiographic indices from 16-week-old mice.

Parameter	WT + Chow	*Dsg2*^mut/mut^ + Chow	*Dsg2^mut/mut^* + HFD
Morphometric			
BW (g)	25.5 ± 1.1	28.6 ± 1.4	31.5 ± 2.5 *
Heart (mg)	113.6 ± 4.6	127.6 ± 3.3	138.1 ± 6.2 *
Liver (mg)	1062 ± 77.3	1404 ± 69.6 *	1171.1 ± 84.6
Spleen (mg)	77.4 ± 5.5	88.0 ± 4.2	97.0 ± 13.3
HW/BW (mg/g)	4.46 ± 0.08	4.51 ± 0.2	4.52 ± 0.4
LW/BW (mg/g)	41.5 ± 2.1	49.1 ± 0.5 *	37.5 ± 2.0 ^†^
Spleen/BW (mg/g)	3.12 ± 0.3	3.15 ± 0.3	3.30 ± 0.6
*n*-values	*10*	*9*	*8*
Echocardiographic			
IVSd (mm)	0.95 ± 0.02	0.96 ± 0.04	0.85 ± 0.1
IVSs (mm)	1.54 ± 0.03	1.47 ± 0.05	0.80 ± 0.07 *^†^
LVIDd (mm)	2.71 ± 0.1	3.15 ± 0.1*	4.07 ± 0.2 *^†^
LVIDs (mm)	1.12 ± 0.09	1.94 ± 0.1 *	3.06 ± 0.2 *^†^
LVPWd (mm)	0.86 ± 0.03	0.96 ± 0.05	0.86 ± 0.07
LVPWs (mm)	1.50 ± 0.4	1.33 ± 0.05	1.08 ± 0.09
LVEF (%)	83.9 ± 1.2	67.0 ± 1.5 *	57.2 ± 2.7 *^†^
LVFS (%)	59.3 ± 2.0	39.7 ± 1.4 *	24.9 ± 1.7 *^†^
RWT (mm)	0.64 ± 0.03	0.64 ± 0.04	0.42 ± 0.04 *^†^
LVM (mg)	74.7 ± 5.1	98.3 ± 5.3 *	159.6 ± 25.6 *^†^
*n*-values	*15*	*15*	*7*
Electrocardiographic			
Heart rate (bpm)	423 ± 16	468 ± 14	479 ± 19
RR-I (ms)	145.6 ± 5.7	130.0 ± 3.8 *	126.5 ± 5.0 *
PR-I (ms)	42.1 ± 1.3	39.0 ± 1.6	39.5 ± 5.9
Pd (ms)	11.3 ± 0.5	8.77 ± 0.5 *	8.18 ± 0.7 *
QRSd (ms)	10.6 ± 0.3	12.5 ± 0.8	11.7 ± 0.6
P-Amp (mV)	0.08 ± 0.01	0.07 ± 0.01	0.04 ± 0.01 *
Q-Amp (mV)	−0.08 ± 0.01	−0.10 ± 0.04	−0.03 ± 0.01
R-Amp (mV)	0.99 ± 0.08	0.59 ± 0.06 *	0.54 ± 0.04 *
S-Amp (mV)	−0.49 ± 0.02	−0.29 ± 0.06	−0.47 ± 0.09
J-wave depression (%)	7.7 ± 0.08 (*n* = 1/13)	18.8 ± 0.1 (*n* = 3/16)	87.5 ± 0.1 *^†^ (*n* = 7/8)
*n*-values	*13*	*14–16*	*8*

WT, wildtype; HFD, high-fat diet; BW, body weight; HW, heart weight; LW, liver weight; IVSd, interventricular septum; d, end-diastole; s, end-systole; LVID, left ventricular internal diameter; LVPW, LV posterior wall; LVEF, LV ejection fraction; LVFS, LV fractional shortening; RWT, relative wall thickness; LVM, LV mass; RR-I, RR interval; PR-I, PR interval; Pd, P-wave duration; QRSd, QRS duration; P-Amp, P-amplitude; R-Amp, R-amplitude; Q-Amp, Q-amplitude; S-Amp, S-amplitude. Data presented as mean ± SEM or percent J-wave depression (via mice that showed J-wave depression divided by total mice within their respective cohort × 100). *n*-values inset; * *p* ≤ 0.05 for any cohort vs. WT + chow; ^†^ *p* ≤ 0.05 for *Dsg2*^mut/mut^ + HFD vs. *Dsg2*^mut/mut^ + chow using one-way ANOVA with Tukey’s post hoc.

## Data Availability

All data used in this study are available online in the Appendix A.

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
