# Peer review of "High-Fat Diet Augments Myocardial Inflammation and Cardiac Dysfunction in Arrhythmogenic Cardiomyopathy"

_nutrients, 2024, doi:10.3390/nu16132087_

Round 1

Reviewer 1 Report

Comments and Suggestions for Authors

In the manuscript “High-Fat Diet Consumption Augments Myocardial Inflammation and Cardiac Dysfunction in a Preclinical Animal Model of Arrhythmogenic Cardiomyopathy”, Ann M. Centner, et al. investigated how a high-fat diet exacerbates myocardial inflammation, cardiac dysfunction, and adverse cardiac remodeling, and explored the correlation between lipid levels and cardiac function. In general, in the study, the authors analyzed the high-fat diet not only exacerbates cardiac dysfunction but also promotes unfavorable cardiac remodeling, and despite the correlation between changes in lipid levels and cardiac function, a high-fat diet is not the initial trigger for cardiac dysfunction in ACM but may act as an environmental factor to accelerate disease progression. However, there are some formatting errors in the tables and images of this article, and appropriate revisions are suggested.

Questions were raised as below and needed to be addressed.

1. The format of "P-value" is not standardized.

2. "±", "" and letters should have space between them

3. 2.4 should be followed by a "."

4. Unit millivolt abbreviation mV can be all lowercase

5. Whether to use "-" or "," between descriptions of neighboring pictures should be standardized, e.g. Figure 1D-E and Figure 2A, B.

6. In the description of the results section, you can add a P-value for significant results.

7. whether the fat content of 60% in this experimental design for a high-fat diet is too high and does not correspond to the current situation described in the introduction.

Reviewer 2 Report

Comments and Suggestions for Authors

The study entitled “High-Fat Diet Consumption Augments Myocardial Inflammation and Cardiac Dysfunction in a Preclinical Animal Model of Arrhythmogenic Cardiomyopathy” represents comprehensive, systematic, and extensive research regarding the role of HFD in cardiac dysfunction. The author used 8-week-old Desmoglein-2 mutant (Dsg2mut/mut) mice, a well-established ACM animal model, and fed them either HFD or standard rodent chow for 8 weeks. Echocardiography, electrocardiography, lipid profiles, inflammatory markers, and myocardial fibrosis were assessed. In conclusion, HFD not only worsened cardiac dysfunction but also promoted adverse cardiac remodeling in this ACM mouse model. The positive correlation between HDL levels and ejection fraction suggests a potential protective effect, while the elevated inflammatory adipokines indicate that diet may be a modifiable environmental factor influencing ACM disease progression. Further investigation is warranted to elucidate the underlying mechanisms. While the study focus is of interest, the manuscript should be improved.

I consider the minor revisions should be done.

Please consider rephrasing the title. It should be shorter if it is possible.

English language, grammar, and typographic corrections should be performed.

The Introduction section: This section is well constructed and the aim was clear.

The described mechanisms responsible for the effects of HFD are quite superficial and I believe it could be improved.

Conclusions need to be more focused and place findings in context.

Comments on the Quality of English Language

English language, grammar, and typographic corrections should be performed.

Round 2

Reviewer 1 Report

Comments and Suggestions for Authors

Accepted